# Suicidality in Borderline Personality Disorder

**DOI:** 10.3390/medicina55060223

**Published:** 2019-05-28

**Authors:** Joel Paris

**Affiliations:** Institute of Community and Family Psychiatry, McGill University, 4333 chemin de la cote ste. Catherine, Montreal, QC H3T1E4, Canada

**Keywords:** suicide suicidality personality disorder, borderline personality disorder, self-harm

## Abstract

Borderline personality disorder (BPD) is associated with suicidal behaviors and self-harm. Up to 10% of BPD patients will die by suicide. However, no research data support the effectiveness of suicide prevention in this disorder, and hospitalization has not been shown to be useful. The most evidence-based treatment methods for BPD are specifically designed psychotherapies.

## 1. Introduction

This review is based on a search with the key words “borderline personality disorder” and “suicide or suicidality” in MEDLINE and PsycINFO of all articles since 2000 (along with some important older articles). This review focuses on articles that are most relevant to the following questions:

(1) What suicidal behaviors are seen in patients with borderline personality disorder (BPD), and what is their motivation?

(2) What is the risk for death by suicide in BPD?

(3) Is there evidence for the value of hospitalizing suicidal BPD patients? 

(4) What are the most evidence-based treatments for suicidal BPD patients?

## 2. Suicidal Behaviors in BPD

As defined by the DSM-5 [1] and by the ICD-11 [2], personality disorders (PDs) are characterized by abnormal patterns of inner experience and behavior. These affect cognition, emotion, interpersonal functioning, impulse control, are inflexible and pervasive, lead to clinically significant distress or impairment, are stable and of long duration, and have an onset in adolescence or early adulthood. Personality disorders are common in practice, and can be found, above and beyond other diagnoses, in up to 45% of all outpatients [3]. Of these, BPD is by far the most researched category due to the clinical challenges it presents.

BPD is associated with a wide range of psychopathology, including unstable mood, impulsive behaviors, as well as unstable interpersonal relationships [1]. BPD patients have a mean of three lifetime suicide attempts, mostly by overdose [4]. 

Self-harm behaviors (i.e., non-suicidal self-injury) (NSSI) [5], are also common in BPD. NSSI usually presents as superficial cuts to the wrists and arms. However, NSSI is not suicidal in intent; BPD patients have problems with emotional regulation and cut themselves addictively to reduce painful inner states [6]. Cutting relieves emotional tension, but does not reflect a wish to die [7]. 

While drug overdoses can sometimes be life threatening, these behaviors vary greatly in nature and intent. They usually occur following stressful life events, and patients describe their motivation as a wish to escape [7]. Most incidents reflect ambivalent motivation, involving small quantities of medication and/or calling significant others for help. Even when potentially fatal overdoses occur, patients often contact people who are in a position to intervene. 

It has long been established that suicide attempters and completers are separate but overlapping populations [8]. In a large-scale follow-up of attempters seen in an emergency room (ER), only about 3% eventually died by suicide [9]. Most repetitive attempts occur in young women, and decrease with time [10]. 

## 3. Death by Suicide in BPD Patients

Follow-back research has found that suicide occurs in up to 10% of BPD cases [11,12]. However, lower rates (3%–6%) have been reported in prospectively followed cohorts [13,14,15]. These discrepancies could reflect less-severe suicidality in patients who agree to be followed in research studies.

By and large, suicides in BPD occur later in the course of the illness and follow long courses of unsuccessful treatment [16]. A 15-year follow-back study found that the mean age at suicide to be 30 [12], while a 27-year follow-up reported a mean age of 37, with a standard deviation of 10 [11]. Thus, patients are not at their highest risk of suicide when they are young and frequent visitors to the ER. 

Even so, patients with BPD do kill themselves. Psychological autopsy methods, which involve post-mortem interviews with families, have examined the frequency of this diagnosis in death by suicide [17,18,19]. In these studies, PDs were present in about half of the cases under the age of 35, with BPD being the most common category.

A meta-analysis by Pompili et al. [20] noted that suicidality is “more alarming” in young people, with a high level of suicidal behaviors. Yet the great majority of BPD patients eventually improve over time [16], and those who die by suicide tend to be those who fail to recover.

Males with BPD have a different pattern. Nearly a third of youth suicides, most of whom are male, can be diagnosed with BPD by psychological autopsy [17]. Other studies of BPD patients who died by suicide also show a preponderance of males [21]. Few of these patients were in treatment at the time of their death.

## 4. Hospitalization for Suicidality in BPD Patients

Many BPD patients have courses of treatment marked by multiple failed suicide attempts. These treatments are associated with multiple trials of psychotherapy, multiple prescriptions, repeated emergency room visits, and hospitalization for suicide attempts and threats [22]. Yet the research literature on the management of suicidality in BPD does not provide evidence-based guidelines for the prevention of death by suicide [23]. 

It has long been known that clinicians lack algorithms that can predict a fatal outcome in major mental disorders with any accuracy. As summarized in an article in The Lancet [24], in spite of a large body of research on suicide prediction, there is still no effective algorithm that can be used in practice to predict suicide. At best, clinicians are left with guidelines that may be commonsensical, but that lack empirical support.

However, the absence of evidence for prevention has not changed practice in managing BPD. Specifically, many patients are admitted to hospital when they attempt or threaten suicide. It is understandable that clinicians want to err on the safe side, but the absence of controlled data prevents us from concluding that hospital stays offer an effective method of suicide prevention. Admission for suicidal threats was recommended by the American Psychiatric Association guidelines for the treatment of BPD [25]. However, these guidelines, which have never been updated, were based on clinical opinion, and not on data showing that hospital admission has a preventive effect.

Repeated hospitalization for suicidal threats and attempts can also be counter-productive, since it interferes with out-patient treatment, and makes it impossible for patients to remain in the workplace [26,27,28]. It can also lead to a kind of “regression”, with an increase of symptoms based on the behavioral reinforcement of suicidal behavior. Linehan [6] recommends not admitting patients for suicidality beyond an overnight hold in a crisis.

## 5. Evidence-Based Treatments for BPD

We now know that well-structured ambulatory treatment, using methods specifically developed for BPD, is an effective intervention for most BPD patients. The most extensively investigated treatment is dialectical behavior therapy (DBT) [6], whose efficacy has been confirmed by randomized clinical trials [29,30]. The main outcomes are reductions in overdoses, in emergency room visits for suicidality, reduced frequency of self-harm, and reduced hospital admissions. DBT specifically aims to modify the regulation of emotion, teaching patients how to regulate negative emotions in ways other than cutting or overdosing [6]. 

Several other psychotherapy methods have also been tested in randomized clinical trials: mentalization-based therapy [31,32]; transference-focused psychotherapy [33]; schema-focused therapy [34]; and standard cognitive therapy [35]. 

All these methods target the affective instability (emotion dysregulation) that characterizes BPD [6,36], as well as impulsivity [37]. Emotion dysregulation accompanied by a lack of control over impulses makes patients more likely to turn to suicidal actions, and the frequency of suicide attempts is strongly related to these traits [10]. Thus, all methods of psychotherapy attempt to teach patients to stand outside their emotions, to self-reflect before acting on them, and to develop a better understanding of interpersonal conflict.

By and large, all psychological treatments that are well-structured and specifically designed for BPD patients are superior to standard clinical management. A Cochrane report [38] as well as a systematic meta-analysis [39] have summarized this evidence, supporting the conclusion that specific forms of psychotherapy for BPD are efficacious. These methods are usually provided in out-patient settings, and do not require hospital admission. As noted by Zanarini [40], BPD patients need to “get a life”, which means therapists must work actively to involve them with life goals, such as career and social networks.

In contrast, the efficacy of pharmacological agents in BPD is not well established. No clinical trials have documented remission of the disorder with successful drug treatment, and a Cochrane report did not find sufficient evidence to prescribe *any* drug for patients with BPD [41]. Recent research has also shown that anticonvulsant mood stabilizers are not efficacious [42]. There is some evidence for the use of antipsychotics for short periods [43]. A similar conclusion has been reached by the UK National Institute on Clinical Excellence [44]. Unfortunately, it has long been observed that most BPD patients are often on multiple medications, including antidepressants, mood stabilizers, and/or neuroleptics [22], and this practice does not seem to have changed. These interventions do not require hospital admission. There is also no evidence that pharmacological regimes are effective for suicide prevention.

## 6. Implications for Practice

One of the most unique aspects of BPD is chronic suicidal ideation [27]. Patients with mood disorders can be suicidal when depressed, but usually put these ideas aside when they go into remission. In contrast, BPD patients may consider suicide on a daily basis for months to years, and only go into remission much later. Suicidal ideas will vary in intensity over time, waxing when life events are stressful, and waning when they are not.

Yet by themselves, suicidal thoughts are too common to be useful in predicting suicidal actions. But while patients with suicidal behaviors have a statistically higher risk, one cannot predict who is most likely to die by suicide. Deaths by suicide are rare events relative to attempts, which is why large-scale follow-up studies have found that algorithms based on risk factors fail to predict who will die by suicide [45,46]. The problem is false positives (patients who fit a profile but never kill themselves). 

Most patients with BPD, despite having suicidal thoughts for long periods of time and multiple suicide attempts, never kill themselves. Thus, the level of alarm created by patients with BPD who present in clinics and ERs with suicidal ideas is not necessarily justified, even when threats are dramatic or blood-curdling. Clinicians need to work on making these patents more functional, and should not be distracted from these therapeutic tasks by suicidality.

Needless to say, chronic suicidality can be draining for therapists, and no one wants to lose a patient in this way. Yet in BPD, suicidality “goes with the territory” [47], and most patients cannot be treated without accepting a calculated risk [48]. Moreover, recommending ER visits and hospitalization reinforces the very behaviors they are designed to treat [6]. 

Hospitalization has not been supported by evidence, and when suicidality is chronic, admission to hospital provides only temporary relief; most patients continue to have suicidal ideas after discharge. While some research describes intensive treatment in hospital [49], similar programs could be offered on an out-patient basis. To avoid the harm of repetitive admissions, one might prefer day treatment, which offers the advantages of admission (intensive treatment by an experienced team) without its disadvantages, and has some supporting evidence for its efficacy [31]. Unfortunately, day programs usually have waiting lists and are not useful in a crisis.

In practice, BPD patients are commonly held in ERs (or admitted to wards) when they threaten suicide, cut themselves, or overdose. These choices are partly determined by a fear of litigation. However, to minimize the risk of lawsuits, clinicians can ensure careful record keeping, consult frequently with colleagues, and get families involved early on in treatment [50].

Based on current evidence, it is reasonable to conclude that we should treat suicidal patients with BPD on an out-patient basis using specialized forms of psychotherapy. Psychopharmacology remains adjunctive and optional. Hospital admission can be justified by either a near-fatal attempt (requiring a re-evaluation), or a micro-psychotic episode (requiring pharmacological intervention) [27]. Since we have no firm evidence that death by suicide in BPD can be prevented, we should focus on providing existing evidence-based psychological treatments designed for this challenging population.

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
