# Peer review of "Suicidality in Borderline Personality Disorder"

_medicina, 2019, doi:10.3390/medicina55060223_

Reviewer 1 Report

The paper describes the interconnecting lines between Borderline Personality Disorder, Personality Disorders, and suicidality (both attempted suicide and suicide).

The paper thereby offers some arguments why certain people develop stronger death wishes than others (“escape”, population overlap, repetitive character of suicidality, BPD-suicidality comorbidity), and then presents management strategies for suicidality within BPD.

However, these first two sections remain completely descriptive and have no clear question that the review article tries to answer here. This makes these parts purely enumerative and less informative than they could be.

With regard to the final section on implications for practice I was wondering whether everybody would agree with the suggested implication that “the level of alarm created by young women with BPD who present in clinics and ERs with suicidal ideas is not justified, even when threats are dramatic or blood-curdling”. Moreover, why is the focus here particularly on young women? This has not been previously justified. 

 I do not agree with the concluding statement that “based on current evidence, it is clear that we should treat suicidal patients with BPD on an outpatient basis”. If the paper is focused on making this claim, that should guide the whole research process including the reading of the included literature.

Methodologically, there is no information about the timeframe or the selection criteria that have been applied to select the literature basis for this review article. It seems more to be an opinion piece, which is not necessarily a scientific writing style.

Many expressions and word choices seem odd and the manuscript would require a thorough proofreading.

Author Response

Reviewer 1

The paper describes the interconnecting lines between Borderline Personality Disorder, Personality Disorders, and suicidality (both attempted suicide and suicide).

The paper thereby offers some arguments why certain people develop stronger death wishes than others (“escape”, population overlap, repetitive character of suicidality, BPD-suicidality comorbidity), and then presents management strategies for suicidality within BPD.

However, these first two sections remain completely descriptive and have no clear question that the review article tries to answer here. This makes these parts purely enumerative and less informative than they could be.

I have now stated what my questions are at the beginning of the paper, and reorganized the paper along these lines from start of finish

With regard to the final section on implications for practice I was wondering whether everybody would agree with the suggested implication that “the level of alarm created by young women with BPD who present in clinics and ERs with suicidal ideas is not justified, even when threats are dramatic or blood-curdling”. Moreover, why is the focus here particularly on young women? This has not been previously justified. 

I have revised this statement. Most of these patients are in fact, young and women, but the point is that research does not show that death by suicide is a high risk in BPD patients seen in emergency settings.

I do not agree with the concluding statement that “based on current evidence, it is clear that we should treat suicidal patients with BPD on an outpatient basis”. If the paper is focused on making this claim, that should guide the whole research process including the reading of the included literature.

We do not agree! My point is that there is no evidence that hospitalization prevents suicide. A null hypothesis has to hold until more data emerges. This is the central argument of the section on management.

Methodologically, there is no information about the timeframe or the selection criteria that have been applied to select the literature basis for this review article. It seems more to be an opinion piece, which is not necessarily a scientific writing style.

I agree, and have added a statement at the beginning about how I selected references.

Many expressions and word choices seem odd and the manuscript would require a thorough proofreading.

I have done this.

 Reviewer 2 Report

The manuscript is important toward encouraging an increase in suicide prevention, especially those with BPD. However, to be honest, I am disappointed in what appears to be a lack of energy dedicated to finding the most current, accurate work to support the statements wiithin the manuscript. Sources are far too outdated which lessens the credibility.

Second paragraph in the intro. What is meant by “clinically important?”

These “atients.” Does the author mean, “patients?”

Paragraph 3 of the introduction is very choppy in its communication. There are sentences presented with little to know elaboration.

The sentence that begins with “The distinction between NSSI…” is unnecessarily long and goes into terminology that doesn’t need to be presented. Specifically, the authors should state their term used for the behaviors and make it clear that is the way they are operationalizing it within the article.

The authors cite research that is far too old to back up their claims. While historical works are important, one should always look into more recent works. A number of the articles within the manuscript are at least 10 – 15 years old, which is unacceptable when dealing with a current issue. I am not referring to the method section. Rather, the dated works are used to support their claims within the introduction. They must be updated.

This statement is made within the article: “The research literature on the management of suicidality in BPD does not provide clear guidelines for prevention” (p. 2). It is false. By extension, a simple google search found the following studies/prevention recommendations on suicide prevention among patients with BPD:

https://www.ncbi.nlm.nih.gov/pubmed/27000268

https://www.bphope.com/straight-talk-about-suicide/

The connection between BPD and suicide is well documented. As such, there are researchers and clinicians devoted to suicide prevention amon those struggling with BPD. To state otherwise is incorrect.

Within the Implications for Practice section, again, the sources are outdated. The study would be more credible with current information.

Author Response

Reviewer 2:

Comments and Suggestions for Authors

The manuscript is important toward encouraging an increase in suicide prevention, especially those with BPD. However, to be honest, I am disappointed in what appears to be a lack of energy dedicated to finding the most current, accurate work to support the statements within the manuscript. Sources are far too outdated which lessens the credibility.

It is true that most of the research on this subject was published at least 10 years ago. These remain the most important studies  However, I have updated the references to include more recent articles.

Second paragraph in the intro. What is meant by “clinically important?”

I have rephrased this paragraph

These “atients.” Does the author mean, “patients?”

Yes, typo corrected.

Paragraph 3 of the introduction is very choppy in its communication. There are sentences presented with little to know elaboration.

I have rewritten this paragraph.

The sentence that begins with “The distinction between NSSI…” is unnecessarily long and goes into terminology that doesn’t need to be presented. Specifically, the authors should state their term used for the behaviors and make it clear that is the way they are operationalizing it within the article.

I have introduced the term “non-suicidal self injury” and a reference that defines the construct.

The authors cite research that is far too old to back up their claims. While historical works are important, one should always look into more recent works. A number of the articles within the manuscript are at least 10 – 15 years old, which is unacceptable when dealing with a current issue. I am not referring to the method section. Rather, the dated works are used to support their claims within the introduction. They must be updated.

As noted above, I have quoted the most important research, much of which I agree is not recent. But if the best studies are older, they need to be quoted. After all, this a slowly evolving field!  I had another look at Medline and Psycinfo, but there are only a few recent papers that significantly contribute to the field. Nonetheless, I have updated the references with these more recent articles.

This statement is made within the article: “The research literature on the management of suicidality in BPD does not provide clear guidelines for prevention” (p. 2). It is false. By extension, a simple google search found the following studies/prevention recommendations on suicide prevention among patients with BPD:

https://www.ncbi.nlm.nih.gov/pubmed/27000268 

But this article refers to lithium in bipolar disorder.

https://www.bphope.com/straight-talk-about-suicide/ 

This article is written for the general public, and lacks an evidence base..

While some have published guidelines for managing chronic suicidality, suicidologlsts generally agree that hard evidence for prevention is lacking. I have added a reference to an important review in Lxancet by Turecki and Brent.I  therefore disagree with the view that my conclusion is “false”.

The connection between BPD and suicide is well documented. As such, there are researchers and clinicians devoted to suicide prevention among those struggling with BPD. To state otherwise is incorrect.

I do not doubt the connection with suicide, but my point concerns the lack of empirical evidence for prevention. I have expanded the argument to explain that while there is good evidence for treatment efficacy and effectiveness, using hospitalization to prevent suicide in these patients is not evidence-based.

Within the Implications for Practice section, again, the sources are outdated. The study would be more credible with current information.

As noted above,I have updated my references.

 Round  2

Reviewer 1 Report

All of my previous comments were adressed.

Reviewer 2 Report

In terms of the lithium article, the point is that there is work being done to examine what works as a form of prevention among suicidal patients with BPD, which is a testament to the need for this type of research. In truth, the article is also relevant in terms of its statement regarding suicidality in patients with BPD. However, the updates added by the author will suffice.